# Lipophilic Substances of the Leaves and Inflorescences of *Centaurea scabiosa* L.: Their Composition and Activity Against the Main Protease of SARS-CoV-2

**DOI:** 10.3390/molecules30234568

**Published:** 2025-11-27

**Authors:** Tatiana P. Kukina, Ivan A. Elshin, Maria K. Marenina, Evgeniia A. Kolosova, Yulia V. Meshkova, Ol’ga I. Sal’nikova, Svetlana V. Belenkaya, Ekaterina A. Volosnikova, Mikhail V. Khvostov, Dmitry N. Shcherbakov

**Affiliations:** 1Vorozhtsov Novosibirsk Institute of Organic Chemistry, Siberian Branch of the Russian Academy of Sciences, 630090 Novosibirsk, Russia; fominamk@gmail.com (M.K.M.); meshkova@nioch.nsc.ru (Y.V.M.); olga@nioch.nsc.ru (O.I.S.); belenkaya.sveta@gmail.com (S.V.B.); khvostov@nioch.nsc.ru (M.V.K.); dnshcherbakov@gmail.com (D.N.S.); 2Department of Natural Sciences, Novosibirsk State University, 630090 Novosibirsk, Russia; lanosterol@yandex.ru; 3State Research Center of Virology and Biotechnology VECTOR, Rospotrebnadzor, 630559 Koltsovo, Russia; kurchanovaea@gmail.com (E.A.K.); volosnikova_ea@vector.nsc.ru (E.A.V.); 4Research Institute of Biological Medicine Center for Recombinant Technologies, Altay State University, 656049 Barnaul, Russia

**Keywords:** *Centaurea scabiosa* L., cornflowers, greater knapweed, GC-MS, triterpenoids, sterols, 3CL protease, SARS-CoV-2

## Abstract

The composition of the lipophilic components of *Centaurea scabiosa* L. has been studied. The raw material was subjected to extraction with hexane and methyl tert-butyl ether (MTBE) using both exhaustive and sequential schemes for a detailed characterization. The resulting extracts were fractionated into acidic and neutral components via treatment with alkali solutions. The acidic compounds were converted into methyl esters for subsequent gas chromatography–mass spectrometry (GC-MS) analysis, while the neutral unsaponifiable fractions were separated into groups of different polarities using column chromatography on silica gel. This approach enabled the identification of a complex profile of lipophilic substances. In the acidic fractions, aliphatic acids with chain lengths from C_10_ to C_32_, including unsaturated variants, were characterized. The neutral fractions revealed over compounds, encompassing n-alkanes, substantial levels of the unsaturated branched hydrocarbon squalene, and a diverse array of oxygenated terpenoids. The latter were mainly represented by highly active triterpene alcohols and ketones belonging to the ursane, oleanane, lupane, and cycloartane types. The sterol composition was dominated by β-sitosterol and accompanied by cholesterol, campesterol, stigmasterol, stigmast-7-en-3-β-ol, fucosterol, and stigmastan-3-β-ol. Bioactivity screening demonstrated that several of the obtained lipophilic extracts, particularly those of lower polarity, exhibited high inhibitory activity against the main protease of SARS-CoV-2, underscoring the potential of *C. scabiosa* as a valuable source of anti-coronavirus agents.

## 1. Introduction

The genus *Centaurea* belongs to the Asteraceae (Compositae) family and comprises 400–700 species of annual, biennial, and perennial herbaceous plants distributed throughout Europe and Asia [1,2,3,4]. Many of these species have been used in traditional medicine to treat various diseases and conditions [5,6,7,8]. In folk medicine, cornflower species are used as remedies, primarily utilizing the flowers, stems, and leaves. Infusions are employed for jaundice, rashes, and dropsy and as an analgesic. For instance, *Centaurea cyanus* and *Centaurea scabiosa* are used as diuretics and tonics in Scottish medicine [9]. In Turkish folk medicine, *C. pulchella*, *C. drabifolia*, and *C. solstitialis* are used to treat abscesses, hemorrhoids, peptic ulcers, and the common cold [7,8]. Due to their potential applications, several studies have focused on the secondary metabolites and biological activities of *Centaurea* species [5,6,7,8,9,10,11,12]. In Turkey, this genus is represented by a very large number of species (more than 180). Many of them are endemic. Approximately 22 species found in Russian Federation [3,4], with about 10 species found in Siberia [4].

A detailed review [13] describes the chemical composition of *Centaurea* species, with sesquiterpene lactones as compounds being of greatest interest due to antiviral and antifungal activity [14,15,16,17,18,19,20,21]. The literature data provide the composition of essential oils and the antibacterial activity of more than 70 cornflower species, with over 400 different compounds identified [22,23,24,25,26,27,28,29,30,31,32,33,34,35]. However, the lipophilic components of genus *Centaurea* (knapweed) have not been sufficiently studied. Some researchers have identified aliphatic (fatty) acids [9,36,37,38,39,40,41,42,43], as well as some sterols and triterpenoids [5,6,9,13,40,41,42,43]. Greater knapweed *Centaurea scabiosa* L. is particularly noteworthy due to its wide distribution, high biomass accumulation during the vegetation period, and successful attempts at the introduction and cultivation of callus culture [44]. *C. scabiosa* L. has attracted significant research interest in several countries because of its pronounced hepatoprotective and antioxidant properties [5,6,9,25,26,45]. Extracts from different parts of the plant exhibit anti-inflammatory, diuretic, astringent, and sedative effects. In folk medicine, it is used for menorrhagia, malaria, stroke, epilepsy, gastralgia, dermatitis, and scrofula [5,6]. Its anthelminthic and anticonvulsant properties have also been studied [46,47]. Several sesquiterpene lactones [14,15] and phenolic compounds, such as flavonoids, hydroxycinnamic acids, and coumarins, have been well studied [6,12]. Carbohydrates, polyacetylene substances, vitamins, and some lipids have also been found [5,6,9,13,48,49,50]. However, hydrocarbons, aliphatic aldehydes, ketones, and alcohols remain largely unstudied, except for n-alkanes with chain lengths of 23–29, 31, and 33 carbon atoms [9]. Data on triterpenoids and sterols are limited to β- and α-amyrins and β-sitosterol [9,13,42,43]. A detailed study of the inflorescences of some species of cornflowers, undertaken by our colleagues from the Kazan Scientific Center, contains information on γ-sitosterol; however, the reliability of the mass spectrum interpretation, which almost completely coincides with the spectrum of β-sitosterol, is questionable [9]. Furthermore, information on the metabolite composition of the plant’s roots is absent from the literature [13,42].

The emergence of the COVID-19 pandemic has intensified the search for novel antiviral agents, including those of plant origin [51,52,53], particularly those targeting essential viral enzymes. The main protease (3CL or Mpro) of SARS-CoV-2 is a pivotal non-structural protein responsible for cleaving the viral polyprotein. From a structural perspective, the 3CL protease is a homodimeric cysteine protease with a substrate-binding pocket that is highly conserved among coronaviruses yet distinct from human proteases, making it an ideal target for selective inhibition Its pivotal role in processing the viral polyprotein into functional units makes it a critical target for interrupting the viral life cycle [54]. Inhibiting this enzyme effectively halts viral replication [55].

Our research group has a continuing interest in exploring Siberian plants as potential sources of anti-SARS-CoV-2 agents. In this context, our earlier studies on *Caragana jubata* and *Rhododendron adamsii* revealed that their lipophilic extracts can exhibit inhibitory activity against the main protease [56,57], which we tentatively attributed to their terpenoid and sterol content. This observation, coupled with our parallel research into the 3CL inhibitory activity of specific natural product derivatives [58,59], led us to speculate that other plant species rich in similar lipophilic metabolites might share this property. Therefore, we sought to evaluate *Centaurea scabiosa* L., a plant documented to contain diverse terpenoids and used in traditional medicine, to determine whether its lipophilic extracts also demonstrate activity against this key viral enzyme. Therefore, the aim of our work was to study in detail the composition of the poorly investigated lipophilic components of the aerial parts of *C. scabiosa* and to test the antiviral activity of its extracts and fractions from this raw material against the main protease of SARS-CoV-2.

## 2. Results

### 2.1. Chemical Composition of Lipophilic Substances of Centaurea scabiosa L.

The lipophilic extracts from the aerial parts (leaves and inflorescences) of *Centaurea scabiosa* L. (Figure 1), obtained via exhaustive and sequential extraction with hexane and methyl tert-butyl ether (MTBE), were subjected to detailed analysis. The initial focus was on the unsaponifiable residues (URs), which represent a previously unexplored chemical profile of this plant species. The subsequent chromatographic fractionation of the combined URs on a silica gel column successfully separated the complex mixture into distinct compound classes based on their polarity. This process yielded four fractions: hydrocarbons, aldehydes and ketones, aliphatic and terpenic alcohols, and a polar fraction of sterols and triterpenoids. The composition of each fraction was identified using GC-MS, and the concentration of each component was quantified relative to the dry mass of the raw plant material. The complete quantitative data for all identified components in the unsaponifiable residue, including their retention indices and concentrations expressed in mg per 100 g of dry raw material (mg%), are presented in Table 1, Appendix A, Table 2 and Appendix A for inflorescences and leaves, respectively. This detailed profiling revealed a remarkable diversity of over 100 terpene and aliphatic compounds previously unreported for this species.

The content of acid constituents is presented in both Appendix A in mg%, for ease of comparison of different types of raw materials and different extraction schemes. The results of GC-MS analysis allowed us to identify some low-polarity compounds that have not been discovered before in this kind of raw plant material, in particular aliphatic ketones, polyunsaturated hydrocarbons (including squalene), saturated branched hydrocarbons, polyunsaturated aliphatic alcohols, and aliphatic aldehydes.

More than 100 terpene and aliphatic compounds were identified in the URs (Table 1 and Table 2, Appendix A). We detected aliphatic hydrocarbons with chain lengths of 10 to 37 carbon atoms, including branched structures and the bioactive compound squalene (Appendix A). By comparing the GC-MS spectra with databases, we also identified eight aliphatic aldehydes. It is noteworthy that aliphatic aldehydes are almost completely extracted by hexane. The structure of 10 aliphatic ketones was established, i.e., among them, 7 were determined to be aliphatic with a keto group mainly in the second position. A total of 26 aliphatic alcohols were identified in inflorescences and stems with leaves. 1-Alkanes were the main constituents, but components with a hydroxyl group in 10- and 9-position have also been detected. The major sterol component (β-sitosterol) is accompanied by cholesterol, campesterol, stigmasterol, stigmast-7-en-3-β-ol, and fucosterol. In addition to sterols revealed previously, highly active triterpene alcohols were detected, represented mainly by oleanane and ursane derivatives. In particular, triterpene alcohols were identified, such as α- and β-amyrins and obtusifoliol; in addition, the content of triterpene alcohols was higher than the concentrations of the corresponding ketones. Both inflorescences and leaves with stems contain significant amounts of cycloartane compounds, especially diols (Table 1 and Table 2).

The data on the concentrations of components in the acid parts of extracts (mg%) calculated from the mass of the raw material, as well as retention time values for the detected acid components, are presented in Appendix A. Appendix A includes the content of components in the acid part of lipophilic extracts of *C. scabiosa* inflorescences. Appendix A deals with the content of components in the acid part of lipophilic extracts of *C. scabiosa* leaves with stems. By comparing the GC-MS spectra with databases, we identified aliphatic acids with chain lengths of 7 to 32 carbon atoms, including unsaturated and branched ones, the compounds of cinnamonic and benzoic series, acids with a cyclopropane fragment, and epoxy, hydroxyl, and keto compounds. Isomers of conjugated octadecatrienoic acid were also detected.

### 2.2. Inhibition of the Main Protease of SARS-CoV-2 by Extracts

The antiviral activity of *Centaurea scabiosa* L. extracts was evaluated by testing their ability to inhibit the SARS-CoV-2 3CL protease; the results are summarized in Table 3. Using nirmatrelvir as a positive control (half-inhibitory concentration (IC_50_) = 0.13 ± 0.03 µM), it was demonstrated that lipophilic extracts obtained with low-polarity solvents exhibited the highest inhibitory activity. No significant difference was observed between the raw materials, as both leaf and inflorescence extracts displayed IC_50_ values below 0.6 mg/mL. The most potent activity was found in the hexane extracts of inflorescences, with IC_50_ values of 0.15 and 0.17 mg/mL. In contrast, ethanol–water (Et:H_2_O) and aqueous (H_2_O) extracts showed markedly lower activity or produced interfering fluorescence.

The samples of culture fluid after growing *Rhizobium radiobacter* were also analyzed. However, they did not show any activity.

## 3. Discussion

### 3.1. Comprehensive Profiling of Lipophilic Diversity in C. scabiosa

This study provides, for the first time, a comprehensive characterization of the lipophilic components from the aerial parts of *Centaurea scabiosa* L., successfully addressing a significant knowledge gap. An analysis of literature data indicates that research on the lipophilic components of various *Centaurea* species has primarily focused on identifying essential oil constituents, the yield of which typically does not exceed 0.5% of the raw material weight. Extraction with low-polarity solvents yields 3 to 15% of the extract, depending on the plant species. Our work goes beyond the previously reported fragmentary data on n-alkanes and major triterpenoids, revealing a broad chemical diversity encompassing over 100 compounds, many of which are reported for this species for the first time.

Our analytical approach, involving sequential extraction with hexane and MTBE followed by the detailed fractionation of the unsaponifiable residue, proved highly suitable for deconvoluting this complex metabolic profile. For comparison, exhaustive extraction with MTBE was also performed. This scheme can easily be extended to include more polar solvents to target metabolites insoluble in low-polarity solvents.

The chemical diversity uncovered was remarkable. The hydrocarbon fraction was not limited to the previously known C_23_–C_33_ range [9] but included a homologous series from C_10_ to C_37_. A pronounced predominance of odd-numbered carbon homologs, with n-nonacosane (C_29_) and n-hentriacontane (C_31_) being dominant, represents a classic biosynthetic signature. The profile of oxygenated aliphatic compounds revealed a high level of complexity. A series of long-chain aldehydes (C_21_–C_30_) with a relatively high content of odd-numbered components suggests their formation via the enzymatic decarboxylation of fatty acids. The composition of aliphatic ketones was particularly indicative. While hexahydrofarnesylacetone could be a degradation product, its high concentration indicates significant phytol turnover. More importantly, the identification of a homologous series of long-chain 2-alkanones and, notably, ketones with a mid-chain carbonyl group, such as 9-heptacosanone and 10-nonacosanone, may point to a specific and largely unexplored ketoreductase activity in *C. scabiosa*.

The expansion of the known triterpenoid profile is possibly the most significant contribution of this work. We not only confirmed the presence of α- and β-amyrin but also quantified them, showing a clear predominance of the reduced alcohol forms over their oxidized analogs (α- and β-amyrenone). This indicates an active biosynthetic pathway leading to biologically active pentacyclic triterpene alcohols. The most remarkable discovery in this class was the identification of a rich and diverse set of cycloartane-type compounds. The presence of cycloartenol and 24-methylenecycloartanol is fundamental, as cycloartane is the first cyclized precursor in phytosterol biosynthesis. Moreover, the detection of significant amounts of cycloartane diols, such as isomers of cycloart-23-en-3,25-diol and cycloart-25-en-3,24-diol, is exceptional. Their abundance suggests that C. scabiosa possesses an active branch of the triterpenoid pathway specializing in the modification of the cycloartane skeleton, which may represent a chemotaxonomic marker for this species.

### 3.2. Organ-Specific Metabolic Differences and Their Implications

The unsaponifiable residues from inflorescences and leaves showed both qualitative and significant quantitative differences, providing direct insight into organ-specific biosynthesis. The significantly higher total alkane content in the inflorescences suggests their specialized ecological role, possibly related to protecting reproductive structures from desiccation or pathogens.

While the dominance of β-sitosterol is consistent with the literature data [9,13,42], our study shows its significant accumulation in leaves (up to 123.96 mg%), consistent with its role as a key membrane component in photosynthetic tissues. A high phytol concentration was expected, but its concentration in the inflorescences was found to be comparable to that in the green parts of the plant. The fraction of aliphatic alcohols was one of the most abundant, with a key finding being that secondary alcohols, particularly 10-nonacosanol, are dominant components of the hexane extracts (reaching up to ~48.6 mg% in both organs). In many plants, primary alcohols predominate in cuticular waxes. The significant co-occurrence and abundance of secondary alcohols in *C. scabiosa* across both organs likely indicates a different, constitutive regulatory mechanism in its wax biosynthesis.

Notably, some metabolites with a hydroxyl function were largely extracted with hexane, while others required the more polar MTBE. This most likely indicates that the former are primarily present in the raw material as esters with aliphatic acids, while the latter are present as free alcohols, a distinction that also appears to be organ-independent for the major components.

### 3.3. Correlation Between Chemical Composition and Bioactivity

An important part of this study was the detection of activity of the lipophilic extracts against the SARS-CoV-2 main protease, particularly in those obtained with hexane from inflorescences (IC_50_: 0.15–0.17 mg/mL). The activity of the non-polar extracts suggests that the inhibitory effect is due to the plant’s lipophilic components. Given the literature data on the antiviral properties of natural compounds [16,18,52,53,54,55,56,57,58,59,60,61], it can be assumed that the components we identified may contribute synergistically to protease inhibition.

This is especially true for triterpene compounds, which, together with sterols, constitute a significant portion of the unsaponifiable residues of the studied lipophilic extracts. Triterpenoids of the ursane, oleanane, and cycloartane types, such as α- and β-amyrin, cycloartenol, 24-methylenecycloartanol, and the corresponding diols, detected in significant amounts, are well known for their antiviral activity. Their mechanism often involves the inhibition of viral enzymes, including proteases [52,53,56,57,58,59,60,61,62,63,64,65]. This conclusion is further supported by our previous research on synthetic triterpenoid derivatives, where amides of corosolic and acetylglycyrrhetinic acids, along with ursane triterpenoid hybrids incorporating various heterocyclic moieties, demonstrated potent protease inhibition with IC_50_ values ranging from 8 to 125 µM [58,66].

Furthermore, other identified compound classes are known for their biological potential. The aliphatic aldehydes and ketones, which we describe for the first time in *C. scabiosa*, also possess documented antiviral activity [67]. Similarly, the diverse profile of aliphatic acids and acids of the benzoic and cinnamic series, identified in the acid fractions, may contribute to the overall bioactivity of the plant extracts [68]. These compounds may enhance the inhibitory effect through synergistic interactions or by modulating the solubility and bioavailability of the terpenoids.

In conclusion, the activity of *C. scabiosa* extracts against SARS-CoV-2 is likely not due to a single compound but is the result of the combined action of a spectrum of lipophilic metabolites. The most active extracts, being a rich source of triterpenoids, sterols, and other non-polar biologically active molecules, position *C. scabiosa* as a promising subject of further phytochemical and pharmacological research.

## 4. Materials and Methods

### 4.1. Plant Material

The plant material was collected from the Kosikhinsky District of the Altai Territory. The collection was carried out during the flowering and initial fruiting phase of *Centaurea scabiosa* L. in August of 2020 and August of 2024. It is this phase during which the plant accumulates the largest biomass, in the absence of dead fragments in the aerial parts.

Whole, healthy, but undamaged plants were collected. It was also necessary to take into account the purity of the plant material. If necessary, the dust on aboveground parts, namely the stems, leaves, and inflorescences, was cleaned. The aerial parts including inflorescences and leaves of *Centaurea scabiosa* L. species were used as the samples. The samples of raw material were divided into inflorescences and stems with leaves and dried at room temperature indoors with no exposure to direct sunlight. The authenticity of the raw material was confirmed by Shcherbakov D. N., PhD in Biology.

### 4.2. Preparation of Centaurea L. Extracts

A weighted portion of the raw material was ground in a screw crusher and sieved through a 2 mm mesh. Then, the portion of the raw material, 50 g in mass, was loaded into the Soxhlet extractor and extracted with hexane for 20 h (3 × 7 h). After extraction with hexane, the raw material without unloading was extracted with methyl-tert-butyl ether (MTBE) (PanReac AppliChem, Darmstadt, Germany) for 20 h (3 × 7 h) (sequential extraction). The yield of hexane extract was 1.1 wt% of the raw material mass, MTBE extract—0.5 wt%. For exhaustive extraction, a 500 g portion of raw material was loaded into the Soxhlet apparatus and extracted with MTBE for 20 h (3 × 7 h). The yield was 1.6%. Extracts obtained by high-polarity solvents: a 30:70 water–ethanol mixture, a 60:40 water–ethanol mixture, and water were prepared by remaceration at a temperature of 50–55 °C while stirring on a magnetic stirrer with adjustable heating.

### 4.3. Sample Preparation for GC-MS Analysis

Sample preparation for GC-MS analysis included isolation of free acids with an alkaline extractant (2% aqueous solution of NaOH) and hydrolysis of the extract without free acids using a 15% water–ethanol solution of KOH. The solution of the sodium salts of free acids was acidified with a 10% HCl solution, and free acids were extracted four times with MTBE and washed with distilled water to obtain a neutral reaction. The product of alkaline hydrolysis was 4-fold diluted with water by volume and extracted 4 times with MTBE. Then, the MTBE extract was washed with distilled water to obtain a neutral reaction. The aqueous solution of the potassium salts of bound acids was acidified with a 10% HCl solution, and the acids were extracted four times with MTBE and washed with distilled water to obtain a neutral reaction. Three fractions were obtained from each extract: free acids, bound acids, and unsaponifiable residue (UR). The acid components were methylated with diazomethane, while neutral components were analyzed without derivatization. The neutral substances of the UR were subjected to chromatographic separation through a column filled with silica gel using hexane as the elution solvent with diethyl ether content increasing from 0 to 50% (by volume). The fractions were collected into tubes of 12–15 mL by volume. The fractions were brought together according to the results of thin-layer chromatography on Sorbfil and Silufol plates (Sorbpolymer, Krasnodar, Russia). The chromatograms were developed using a mixture of hexane with MTBE in the concentration from 10 to 50% by volume. Detection was carried out with a mixture of vanillin with sulfuric acid and ethanol in a ratio of 1:10:90 (by mass), with subsequent heating. As a result, the concentrates of hydrocarbons, ketones, and aliphatic and terpene alcohols (including sterols and diols) were obtained and analyzed by GC-MS at the Multi-Access Chemical Research Centre SB RAS.

### 4.4. GC-MS Analysis

The spectra were performed in an Agilent Technologies (Santa Clara, CA, USA) instrument with an Agilent 6890N gas chromatograph and an Agilent 5973N (EI, 70 eV) mass-selective detector using an HP-5MS capillary column [diphenyl (5%)-dimethylsiloxane (95%), 30 m × 0.25 mm × 0.25 μm]. The analysis parameters were as follows: He carrier gas at 1 mL/min; a programmed column temperature of 50 °C for 2 min, from 50 °C to 300 °C at 10 °C/min, and of 300 °C for 30 min; a vaporizer temperature of 300 °C; an ion-source temperature of 230 °C; and a scan rate of 2.4 scans/s in a mass range of 30–650 amu. The components were determined using the W8N08 Library and NIST2020 Library of Mass Spectral Data (more than 500,000 compounds). The contents (%) of the compounds were determined from peak areas in chromatograms without using correction coefficients. The percentage of correspondence between separate components and databases was within the range of 75 to 90%.

### 4.5. 3CL Inhibition Assay

The preparation of the main SARS-CoV-2 protease, 3CL, was carried out in a previously obtained transformant *E. coli* strain, which ensures the synthesis of the target protein in a soluble form. The standard cultivation protocol included the addition of the inducer IPTG. The purification of 3CL included cell biomass lysis and ultrasonic disintegration, as well as the purification of the clarified lysate on Ni-Sepharose [69]. The purity of the resulting sample of 3CL was assessed by SDS-PAGE under denaturing conditions according to the Laemmli method. Protein concentration was determined by the Bradford assay [70].

To evaluate the ability of extracts from *Centaurea scabiosa* L. to inhibit 3CL, IC_50_ was used. It was calculated as the concentration that reduced the fluorescence level by 50% compared to the value obtained without the addition of the inhibitor. Fluorescence was observed by an assay involving a synthetic fluorescently labeled peptide substrate of the type (Dabcyl)KTSAVLQ↓SGFRKME(Edans)NH2 (more than 95% purity, CPC Scientific Inc., Hangzhou, China) containing the site digested by 3CL. The signal was recorded on a SuperMax 3100 fluorimeter (“Flash”, Shanghai, China) at 355 and 460 nm for excitation/emission, respectively, in the kinetic scan mode. Reaction mixtures containing Tris-HCl buffer (supplemented with EDTA, NaCl, DTT; pH = 7.3), 3CL (1200 nM), and the tested extracts (from 5 to 0 mg/mL) were prepared and incubated for 30 min in a 384-well plate at 30 °C, then the reaction was triggered by the addition of fluorogenic substrate (10 µM). The inhibitor of 3CL Nirmatrelvir (LEAPChem, Hangzhou, China) was used as a positive control.

The IC_50_ values were determined from at least three independent biological replicates, each comprising two technical replicates. Data are presented as mean ± standard deviation (SD). The four-parameter logistic (4PL) regression model was applied to the combined data from all replicates to calculate the IC_50_ and its 95% confidence interval using GraphPad Prism 9.0.

## 5. Conclusions

The composition of lipophilic constituents in the aerial parts of *Centaurea scabiosa* L. is investigated. This is a widespread plant species with substantial reserves. These lipophilic components have not been studied previously. The fraction of unsaponifiable residue and acid compounds of the hexane and MTBE extracts of *C. scabiosa* L. were analyzed. The comparison of content of inflorescences and leaves with stems was carried out. More than 100 components of unsaponifiable residues and 82 acid components were identified. Most of them have not been detected previously in this plant species. The identified compounds also include 36 hydrocarbons, in particular the bioactive squalene, 8 aliphatic aldehydes, 15 aliphatic and terpene ketones, and 24 aliphatic and 36 terpene alcohols, including sterols and diols. The plant raw material studied herein can be used as a source of its components. The comparative testing of the inhibition of the SARS-CoV-2 main protease by extracts of the inflorescences and aboveground parts of *C. scabiosa* L. obtained using solvents of different polarities was carried out. Hexane, MTBE, a 30:70 water–ethanol mixture, a 60:40 water–ethanol mixture, and water were used as extractants. It was found that the extracts obtained by low-polarity solvents hexane and MTBE show high activity.

## Figures and Tables

**Figure 1 molecules-30-04568-f001:**
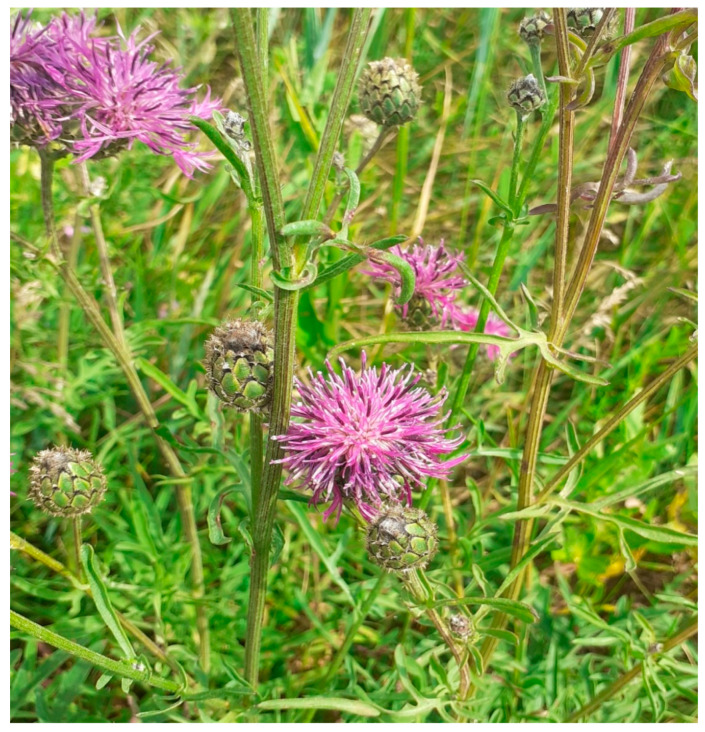
*Centaurea scabiosa* L. Photo from the place of collection of raw material provided by D. N. Shcherbakov.

**Table 1 molecules-30-04568-t001:** The content of components in the unsaponifiable residue of lipophilic extracts of *Centaurea scabiosa* inflorescences calculated for the mass of raw material.

Component	RT (min)	Content, mg%
Hexane Extract	Extract with MTBE After Hexane	MTBE Extract
Aldehydes
Heneicosanal	22.974	0.26	Nd	0.2
Tricosanal	24.620	0.44	Nd	0.38
Tetracosanal	25.392	0.51	Nd	0.49
Pentacosanal	26.143	1.02	Nd	0.96
Hexacosanal	26.865	1.46	Nd	1.42
Heptacosanal	27.601	1.15	Nd	1.12
Octacosanal	28.411	0.77	Nd	0.79
Triacontanal	30.422	0.28	Nd	0.26
Aliphatic ketones
Hexahydrofarnesyl-acetone	18.483	13.29	46.03	39.46
2-Heptadecanone	19.254	1.62	0.09	0.36
2-Nonadecanone	21.210	0.43	Nd	0.44
2-Heneicosanone	23.008	2.26	Nd	2.86
2-Tricosanone	24.668	2.34	Nd	2.33
2-Pentacosanone	25.999	2.2	0.12	2.14
9-Heptacosanone	27.426	5.64	0.23	5.82
2-Heptacosanone	27.394	1.59	0.27	2.69
10-Nonacosanone	29.115	5.04	0.36	5.28
2-Nonacosanone	29.448	0.92	0.05	0.96
Aliphatic alcohols
Decanol	11.618	0.07	1.13	1.09
Dodecanol	14.274	0.18	0.66	0.48
Tetradecanol	16.664	0.28	1.26	0.92
Pentadecanol	17.783	0.24	Nd	0.27
3,7,11,15-Tetramethyl-1-hexadecanol	18.519	0.12	Nd	0.14
Hexadecanol	18.866	1.42	0.66	1.73
Octadecanol	20.822	3.51	1.5	3.85
Isophytol	20.895	0.32	1.01	1.57
Phytol	21.111	20.34	16.67	32.22
Eicosanol	22.656	2.08	0.35	2.97
Geranylgeraniol	23.452	0.48	0.26	0.71
1-Docosanol	24.331	2.41	1.68	3.8
1-Tricosanol	24.375	0.64	0.15	0.79
10-Tricosanol	24.375	1.45	0.34	1.72
1-Tetracosanol	25.876	3.23	0.21	3.31
1-Pentacosanol	26.001	2.25	0.13	2.46
9-Heptacosanol	27.335	12.02	0.13	12.53
1-Hexacosanol	27.535	1.39	0.38	1.48
9-Octacosanol	28.100	1.23	0.14	0.99
10-Nonacosanol	28.916	48.64	0.72	42.14
1-Octacosanol	28.966	1.21	2.53	3.75
10-Triacontanol	30.006	1.55	0.09	0.62
1-Triacontanol	31.183	0.68	1.37	1.98
10-Hentriacontanol	31.204	1.26	0.12	0.99
Terpene ketones
Stigma-3,5-dien-7-one	31.798	0.79	1.03	1.84
Cholesta-3,5-dien-7-one	30.892	0.09	0.26	0.31
β-Amirenone	32.530	0.94	0.12	0.89
α-Amirenone	32.607	0.27	0.31	0.47
22,23-dihydrostigma-3-one	33.112	0.57	0.21	0.75
Terpene alcohols (including sterols and tocopherols)
Spathulenol	15.726	0.75	2.93	2.79
Caryophyllene-α-oxide	15.805	0.71	0.26	1.06
Porosadienol	15.512	0.1	0.89	1.04
Eudesmol, β-	16.585	0.03	0.11	0.13
Caryophylla-3,8(13)-dien-5β-ol, 3*Z*-	16.780	0.11	0.67	0.47
Eudesma-4(15),7-dien-1β-ol	16.975	0.18	0.36	0.42
Cholesterol	29.479	0.97	2.02	2.32
Campesterol	30.677	3.74	9.8	12.29
Stigmasterol	31.089	13.97	20.08	32.75
Obtusifoliol	31.507	0.28	0.19	0.45
β-Sitosterol	31.941	22.42	33.83	62.5
Stigmastanol	32.034	0.23	0.82	1.15
Fucosterol	32.071	1.12	0.38	1.42
24-Methylenelophenol	32.201	0.26	0.16	0.41
β-Amyrin	32.395	4.66	3.54	15.46
Butirospermol	32.482	0.15	0.19	0.29
Stigmast-7-en-3-ol	32.641	9.29	4.28	4.97
Cycloartenol	32.901	3.56	0.58	12.25
α-Amyrin	33.096	11.76	6.9	21.2
Glutinol	33.240	0.16	0.04	0.17
Citrost-7-en-3-ol	33.645	1.12	0.91	1.38
24-Methylenecycloartanol	33.861	1.52	0.34	2.28
Citrostadien-3-ol	33.973	0.51	0.64	0.44
Cycloart-23-ene-3,25-diol	34.061	6.12	0.59	7.57
Simiarenol	34.648	0.21	0.03	0.26
Moretenol	34.540	0.15	3.18	1.84
Taraxasterol	34.756	0.62	0.16	0.85
Cycloart-23-ene-3,25-diol isomer	34.834	0.36	0.55	0.86
Cycloart-25-ene-3,24-diol	36.003	10.01	3.97	13.17
11-Oxo-β-amyrin	36.606	0.42	0.82	1.28
11-Oxo-α-amyrin	37.523	1.12	0.69	1.68
Cycloart-25-ene-3,24-diol, isomer	38.292	10.63	0.13	11.04
Eritrodiol	38.296	0.29	0.08	0.34
Uvaol	39.508	0.17	0.04	0.26
Faradiol	40.042	0.27	0.02	0.32
Betulin	40.114	0.18	0.02	0.21
Nd—not detected; RT—retention time; MTBE—methyl tert-butyl ether. 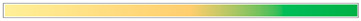 0.01 mg% 70.0 mg%

**Table 2 molecules-30-04568-t002:** The concentrations of components of the unsaponifiable residue of lipophilic extracts of *Centaurea scabiosa* leaves calculated for the mass of raw material.

Component	RT (min)	Content, mg%
Hexane Extract	Extract with MTBE After Hexane	MTBE Extract
Aldehydes
Heneicosanal	22.974	0.23	Nd	0.21
Tricosanal	24.620	0.46	Nd	0.48
Tetracosanal	25.392	0.5	Nd	0.47
Pentacosanal	26.143	1.09	Nd	1.06
Hexacosanal	26.865	1.41	Nd	1.39
Heptacosanal	27.601	1.05	Nd	1.11
Octacosanal	28.41	0.75	Nd	0.77
Triacontanal	30.424	0.31	Nd	0.29
Aliphatic ketones
Hexahydrofarnesyl-acetone	18.483	12.29	36.83	49.26
2-Heptadecanone	19.254	1.09	0.23	1.36
2-Pentacosanone	25.999	0.39	0.12	0.54
9-Heptacosanone	27.426	2.17	Nd	1.98
2-Heptacosanone	27.394	0.15	Nd	0.12
10-Nonacosanone	29.115	5.27	0.45	5.89
2-Nonacosanone	29.448	1.02	Nd	0.9
Aliphatic alcohols
1-Tetradecanol	16.664	0.21	0.26	0.41
1-Pentadecanol	17.783	0.09	0.03	0.11
3,7,11,15-Tetramethyl-1-hexadecanol	18.519	0.08	0.02	0.09
1-Hexadecanol	18.866	0.21	0.36	0.63
9,12-Octadecadien-1-ol	20.560	0.11	0.07	1.18
9-Octadecenol	20.604	0.23	0.11	0.3
1-Octadecanol	20.822	0.51	0.57	1.15
Isophytol	20.895	1.21	1.91	3.37
Phytol	21.111	21.39	20.63	42.27
Eicosanol	22.656	0.48	1.84	2.34
Geranylgeraniol	23.452	0.58	0.12	0.51
1-Docosanol	24.331	2.25	0.29	0.97
1-Tricosanol	24.375	0.44	0.02	0.38
10-Tricosanol	24.375	1.13	0.46	1.51
1-Tetracosanol	25.876	0.46	0.21	1.11
1-Pentacosanol	26.001	0.45	0.12	0.42
9-Heptacosanol	27.335	10.28	Nd	8.53
1-Hexacosanol	27.535	0.39	0.38	0.48
9-Octacosanol	28.100	1.34	Nd	1.04
10-Nonacosanol	28.916	48.66	Nd	43.14
1-Octacosanol	28.966	1.21	3.53	2.75
10-Triacontanol	30.006	1.45	Nd	1.22
1-Triacontanol	31.183	0.6	1.37	0.98
10-Hentriacontanol	31.204	3	Nd	2.03
Terpene ketones
β-Amirenone	32.530	1.04	Nd	1.49
α-Amirenone	32.607	1.67	Nd	2.47
Stigma-3,5-dien-7-one	33.112	5.17	1.32	4.32
Terpene alcohols (including sterols)
Spathulenol	15.726	0.07	0.53	0.73
Caryophyllene-alpha-oxide	15.805	3.76	0.36	4.17
Porosadienol	15.512	0.12	1.26	1.18
Eudesmol, beta-	16.585	0.14	0.05	0.16
Caryophylla-3,8(13)-dien-5β-ol, 3*Z*-	16.780	0.31	0.67	0.88
Eudesma-4(15),7-dien-β-ol	16.975	0.04	0.21	0.24
Cholesterol	29.479	0.78	0.62	0.72
Campesterol	30.677	12.24	3.9	12.09
Stigmasterol	31.089	32.75	10.58	28.59
Obtusifoliol	31.507	4.18	0.19	3.22
β-Sitosterol	31.941	97.26	40.81	123.96
Stigmastanol	32.034	2.03	2.09	11.52
Fucosterol	32.071	0.59	0.18	0.64
24-Methylenelophenol	32.201	5.86	0.16	3.87
β-Amyrin	32.395	17.12	7.84	21.62
Butirospermol	32.482	0.41	0.22	0.59
Stigmast-7-en-3-ol	32.641	1.89	3.76	4.97
Cycloartenol	32.901	3.56	0.58	4.25
α-Amyrin	33.096	21.9	9.67	30.79
Citrost-7-en-3-ol	33.645	2.12	0.32	1.55
24-Methylenecycloartanol	33.861	2.28	0.81	3.58
Citrostadien-3-ol	33.973	17.82	0.6	15.72
Cycloart-23-ene-3,25-diol,	34.061	0.68	14.12	15.54
Moretenol	34.540	5.53	0.89	6.28
Simiarenol	34.648	0.21	0.03	0.26
Taraxasterol	34.756	0.78	0.22	2.08
Cycloart-23-ene-3,25-diol isomer	34.834	0.13	0.76	1.06
Cycloart-25-ene-3,24-diol	36.003	3.21	16.3	23.7
11-Oxo-β-amyrin	36.606	4.43	0.53	4.65
11-Oxo-α-amyrin	37.523	8.18	0.22	7.94
Cycloart-25-ene-3,24-diol isomer	38.292	17.61	0.24	15.43
Eritrodiol	38.296	0.59	0.12	0.46
Uvaol	39.508	0.28	0.11	0.23
Faradiol	40.042	0.49	0.08	0.39
Betulin	40.114	0.22	0.05	0.24
Nd—not detected; RT—retention time; MTBE—methyl tert-butyl ether. 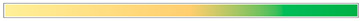 0.01 mg% 70.0 mg%

**Table 3 molecules-30-04568-t003:** The values of the half-inhibitory concentration for extract samples obtained using different solvents.

Code	Raw Material	Extractant	IC_50_, mg/mL
12-7	CentL *	Hexane	0.38 ± 0.08
12-8	CentL	Hexane	0.44 ± 0.09
12-9	CentL	Hexane (22) **	0.43 ± 0.08
12-10	CentL	MTBE (22) **	0.55 ± 0.06
12-11	Centinflor ***	Hexane	0.17 ± 0.02
12-12	Centinflor	Hexane	0.15 ± 0.01
12-13	Centinflor	MTBE/Hexane	****
12-14	Centinflor	MTBE/Hexane	****
12-15	CentL	MTBE/Hexane	0.35 ± 0.06
12-16	CentL	MTBE/Hexane	0.55 ± 0.08
12-17	CentL	MTBE	0.44 ± 0.04
12-18	CentL	MTBE	0.37 ± 0.02
12-19	Centinflor	MTBE	0.27 ± 0.11
12-20	Centinflor	MTBE	0.21 ± 0.05
12-21	Centinflor	Et:H_2_O 70:30	****
12-22	Centinflor	Et:H_2_O 70:30	****
12-23	Centinflor	Et:H_2_O 40:60	****
12-24	Centinflor	Et:H_2_O 40:60	****
12-25	Centinflor	H_2_O	>5
12-26	Centinflor	H_2_O	>5
12-27	CentL	Et:H_2_O 70:30	1.10 ± 0.44
12-28	CentL	Et:H_2_O 70:30	1.08 ± 0.12
12-29	CentL	Et:H_2_O 40:60	2.14 ± 0.38
12-30	CentL	Et:H_2_O 40:60	>5
12-31	CentL	H_2_O	****
12-32	CentL	H_2_O	****
12-33	Cent-roots *****	Hexane	****
12-34	Cent-roots	Hexane	****
14-1	Rhizob ******	H_2_O	>5
14-2	Rhizob	H_2_O	>5

* CentL—aerial part (leaves with stems) of *Centaurea scabiosa* L. ** Samples of extracts prepared in 2022, and the rest were prepared immediately before testing. *** Centinflor—inflorescences of *C. scabiosa* L. **** Unable to determine IC_50_ due to fluorescence extract. ***** Cent-roots—roots of *C. scabiosa* L. ****** *Rhizobium radiobacter*.

## Data Availability

The original contributions presented in the study are included in the article, and further inquiries can be directed to the corresponding author.

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
