# Peer review of "Lipophilic Substances of the Leaves and Inflorescences of Centaurea scabiosa L.: Their Composition and Activity Against the Main Protease of SARS-CoV-2"

_molecules, 2025, doi:10.3390/molecules30234568_

Round 1
Reviewer 1 Report
Comments and Suggestions for Authors
The manuscript addresses a well-targeted and potentially interesting topic with a certain degree of novelty. The study is generally well written, precise, and readable. However, despite the promising title, readers would logically expect more substantial coverage of the SARS-CoV-2-related aspects. The title implies a study directly focused on antiviral activity and protease inhibition, yet the manuscript provides almost no background on SARS-CoV-2, its proteases, or their biological relevance. The Introduction lacks any discussion or literature context regarding viral proteases or SARS-CoV-2 structure, and the Discussion (L258–262) includes only minimal commentary on this topic. This is a major shortcoming of the manuscript, as the connection between C. scabiosa compounds and SARS-CoV-2 inhibition is insufficiently established or discussed.
If direct literature is limited, the authors could still expand the discussion toward related studies on natural products or terpenoids with antiviral activity, to provide stronger scientific context and justify the biological significance of their findings.
Specific Comments and Recommendations:
-
The Latin plant name in the title should be italicized.
-
L16 and throughout the text – Latin names should consistently appear in italics (e.g., L35, 107, 266, 273, 328, 335, 349, etc.). A full revision of the manuscript is recommended to ensure consistency.
-
Table 1 – Include explanations of abbreviations (RT, MTBE) below the table to make it self-contained.
-
Tables 1 and 2 – Adding CAS numbers for identified compounds would increase clarity and reproducibility.
-
Tables 2 and 3 – Again, please provide explanations of all abbreviations used.
-
Table 5 – Remove the dash between the asterisks and the explanatory text in the footnotes for improved readability. Additionally, the symbol “–” used in the table (e.g., rows 7, 8, etc., last column) should be explained in the footnotes.
-
L215 – Possible error in units (“mg%”); please verify and correct.
-
L239 etc. – Ensure correct notation of ICâ‚…â‚€ values, including proper subscript formatting.
-
L281 and L288 – Inconsistent use of dashes versus hyphens for numerical ranges; standardize throughout.
-
L327 – It appears that the term should be “3CL” (main coronavirus protease). Consider rephrasing or clarifying the sentence for precision.
Summary:
The study presents valuable chemical characterization of Centaurea scabiosa extracts and their potential inhibitory activity against SARS-CoV-2 main protease. However, to match the expectations set by the title, the authors should substantially expand the contextual background and discussion concerning SARS-CoV-2 and its protease inhibition mechanisms. After these major conceptual revisions, and minor technical corrections as listed above, the manuscript could be considered for publication.
Author Response
The manuscript addresses a well-targeted and potentially interesting topic with a certain degree of novelty. The study is generally well written, precise, and readable. However, despite the promising title, readers would logically expect more substantial coverage of the SARS-CoV-2-related aspects. The title implies a study directly focused on antiviral activity and protease inhibition, yet the manuscript provides almost no background on SARS-CoV-2, its proteases, or their biological relevance. The Introduction lacks any discussion or literature context regarding viral proteases or SARS-CoV-2 structure, and the Discussion (L258–262) includes only minimal commentary on this topic. This is a major shortcoming of the manuscript, as the connection between C. scabiosa compounds and SARS-CoV-2 inhibition is insufficiently established or discussed.
If direct literature is limited, the authors could still expand the discussion toward related studies on natural products or terpenoids with antiviral activity, to provide stronger scientific context and justify the biological significance of their findings.
Thank you very much for your careful reading of this article. We have endeavored to supplement the text with relevant sections devoted to the main protease in order to expand the information on this aspect of the work.
In Introduction
The emergence of the COVID-19 pandemic has intensified the search for novel antiviral agents, particularly those targeting essential viral enzymes. The main prote-ase (3CL or Mpro) of SARS-CoV-2 is a pivotal non-structural protein responsible for cleaving the viral polyprotein. From a structural perspective, the 3CL protease is a homodimeric cysteine protease with a substrate-binding pocket that is highly con-served among coronaviruses yet distinct from human proteases, making it an ideal target for selective inhibition Its pivotal role in processing the viral polyprotein into functional units makes it a critical target for interrupting the viral life cycle [55]. Inhibiting this enzyme effectively halts viral replication [56].
Our research group has a continuing interest in exploring Siberian plants as potential sources of anti-SARS-CoV-2 agents. In this context, our earlier studies on Caragana jubata and Rhododendron adamsii revealed that their lipophilic extracts can exhibit inhibitory activity against the main protease [57, 58], which we tentatively attributed to their terpenoid and sterol content. This observation, coupled with our parallel re-search into the 3CL inhibitory activity of specific natural product derivatives [59, 60], led us to speculate that other plant species rich in similar lipophilic metabolites might share this property. Therefore, we sought to evaluate Centaurea scabiosa L., a plant documented to contain diverse terpenoids and used in traditional medicine, to determine whether its lipophilic extracts also demonstrate activity against this key viral enzyme. Therefore, the aim of our work was to study in detail the composition of the poorly investigated lipophilic components of the aerial parts of C. scabiosa and to test the antiviral activity of its extracts and fractions from this raw material against the main protease SARS-CoV-2.
In Discussion
An important part of the study was the detection of activity of the lipophilic ex-tracts against the SARS-CoV-2 main protease, particularly in those obtained with hexane from inflorescences (IC50 0.15–0.17 mg/mL). The activity of the non-polar ex-tracts suggests that the inhibitory effect is due to the plant's lipophilic components. Given the literature data on the antiviral properties of natural compounds [16, 18, 53–62], it can be assumed that the components we identified may contribute synergistically to the protease inhibition.
This is especially true for triterpene compounds, which, together with sterols, constitute a significant portion of the unsaponifiable residues of the studied lipophilic extracts. Triterpenoids of the ursane, oleanane, and cycloartane types, such as α- and β-amyrin, cycloartenol, 24-methylenecycloartanol, and the corresponding diols, detected in significant amounts, are well known for their antiviral activity. Their mechanism often involves the inhibition of viral enzymes, including proteases [53,54, 57–66]. This conclusion is further supported by our previous research on synthetic triterpenoid derivatives, where amides of corosolic and acetylglycyrrhetinic acids, along with ursane triterpenoid hybrids incorporating various heterocyclic moieties, demonstrated potent protease inhibition with IC50 values ranging from 8 to 125 µM [59, 67].
Furthermore, other identified compound classes are known for their biological potential. The aliphatic aldehydes and ketones, which we describe for the first time in C. scabiosa, also possess documented antiviral activity [68]. Similarly, the diverse profile of aliphatic acids and acids of the benzoic and cinnamic series, identified in the acid fractions, may contribute to the overall bioactivity of the plant extracts [69]. These compounds may enhance the inhibitory effect through synergistic interactions or by modulating the solubility and bioavailability of the terpenoids.
In conclusion, the activity of C. scabiosa extracts against SARS-CoV-2 is likely not due to a single compound but is the result of the combined action of a spectrum of lipophilic metabolites. The most active extracts, being a rich source of triterpenoids, sterols, and other non-polar biologically active molecules, position C. scabiosa as a promising subject or further phytochemical and pharmacological research.
Specific Comments and Recommendations:
- The Latin plant name in the title should be italicized.
Thank you for your comment, the text of the article has been corrected accordingly.
- L16 and throughout the text – Latin names should consistently appear in italics (e.g., L35, 107, 266, 273, 328, 335, 349, etc.). A full revision of the manuscript is recommended to ensure consistency.
Thank you for your comment, the text of the article has been corrected accordingly.
- Table 1 – Include explanations of abbreviations (RT, MTBE) below the table to make it self-contained.
Thank you for your comment, the text of the article has been corrected accordingly.
- Tables 1 and 2 – Adding CAS numbers for identified compounds would increase clarity and reproducibility.
Thank you for your comment. Using CAS numbers could indeed provide additional clarity. However, on the other hand, it can make tables cumbersome, and using CAS numbers is not a journal requirement. Therefore, we would ask that you leave the text of the article without CAS numbers.
- Tables 2 and 3 – Again, please provide explanations of all abbreviations used.
Thank you for your comment, the text of the article has been corrected accordingly.
- Table 5 – Remove the dash between the asterisks and the explanatory text in the footnotes for improved readability. Additionally, the symbol “–” used in the table (e.g., rows 7, 8, etc., last column) should be explained in the footnotes.
Thank you for your comment, the text of the article has been corrected accordingly.
- L215 – Possible error in units (“mg%”); please verify and correct.
Thank you for your comment, "mg%" is a commonly used unit of measurement for the content of a component in plant material. It represents the number of milligrams of a given component in 100 grams of the original material used for testing. It is convenient for comparing different types of material (including different parts of the plant).
- L239 etc. – Ensure correct notation of ICâ‚…â‚€ values, including proper subscript formatting.
Thank you for your comment, we checked the correct spelling.
- L281 and L288 – Inconsistent use of dashes versus hyphens for numerical ranges; standardize throughout.
Thanks for the comment, we've tried to standardize the use of dashes.
- L327 – It appears that the term should be “3CL” (main coronavirus protease). Consider rephrasing or clarifying the sentence for precision.
Thanks for the point, it is indeed necessary to use the term "3CL".
Summary:
The study presents valuable chemical characterization of Centaurea scabiosa extracts and their potential inhibitory activity against SARS-CoV-2 main protease. However, to match the expectations set by the title, the authors should substantially expand the contextual background and discussion concerning SARS-CoV-2 and its protease inhibition mechanisms. After these major conceptual revisions, and minor technical corrections as listed above, the manuscript could be considered for publication.

Reviewer 2 Report
Comments and Suggestions for Authors
Dear Authors, please see the comments below:
Abstract
-
Seems dense; can be shortened for clarity and impact. Furthermore, the methodology could be simplified.
-
Quantitative details (e.g., “52 to 67 acids”) may be too specific for an abstract; keep key findings concise.
Introduction
-
The introduction is long and occasionally repetitive (e.g., multiple references to folk uses and sesquiterpene lactones).
-
There are several literature citations; please focus on the most relevant 5–6 references per topic
Methods
-
Please avoid repetitive phrasing (e.g., “purity of the plant material” repeated).
-
Include instrument model details (already partially present), ensure consistency throughout.
-
Kindly add clarity on replicates and statistical treatment of ICâ‚…â‚€ data.
Results
-
Please consider moving some tables to Supplementary Materials.
-
Quantitative data could be summarized using charts/heatmaps for easier comparison.
-
Subsections (e.g., hydrocarbons, acids, alcohols) need clearer subheadings and concise interpretation rather than pure listing.
Discussion
-
I suggest groupingthe relevant themes in the discussion.
(1) Lipophilic diversity,
(2) Organ-specific metabolism,
(3) Bioactivity linkages. -
Please strengthen the link to SARS-CoV-2 protease inhibition with literature comparison (e.g., triterpenoids in other species).
Author Response
Dear Authors, please see the comments below:
First of all, the authors would like to thank the reviewer for carefully reading the manuscript and valuable comments.
Abstract
- Seems dense; can be shortened for clarity and impact. Furthermore, the methodology could be simplified.
Thank you for your comment, the authors have shortened the abstract as much as possible.
- Quantitative details (e.g., “52 to 67 acids”) may be too specific for an abstract; keep key findings concise.
Thank you for your comment, the authors have excluded quantitative data from the abstract.
Introduction
- The introduction is long and occasionally repetitive (e.g., multiple references to folk uses and sesquiterpene lactones).
Thanks for the comment, the authors have removed some references to sesquiterpene lactones, especially since they do not fall within the scope of the components we identified. The numerous references are justified by the widespread interest of researchers, as well as the antiviral effect of sesquiterpene lactones. Folk use often points the way to a more in-depth study of plants by chemists, pharmacologists, physicians, and technologists.
- There are several literature citations; please focus on the most relevant 5–6 references per topic
Thanks for the comment, the authors have removed some of the links to studies on the composition of essential oils.
Methods
- Please avoid repetitive phrasing (e.g., “purity of the plant material” repeated).
Thank you for your comment, the authors have tried to remove repetitions in the text.
- Include instrument model details (already partially present), ensure consistency throughout.
Thank you for your comment, the authors have clarified the characteristics of the device and the conditions of its use.
- Kindly add clarity on replicates and statistical treatment of ICâ‚…â‚€ data.
Thank you very much for your comment. The corresponding text has been added to Section 4.5 of the 3CL inhibition assay.
The ICâ‚…â‚€ values were determined from at least three independent biological replicates, each comprising two technical replicates. Data are presented as mean ± standard deviation (SD). The four-parameter logistic (4PL) regression model was applied to the combined data from all replicates to calculate the ICâ‚…â‚€ and its 95% confidence interval using GraphPad Prism 9.0.
Results
- Please consider moving some tables to Supplementary Materials.
Thank you very much for your comment. We've moved some of the tables to the additional resources section.
- Quantitative data could be summarized using charts/heatmaps for easier comparison.
Thank you very much for your comment. We've prepared heatmaps based on the table results.
- Subsections (e.g., hydrocarbons, acids, alcohols) need clearer subheadings and concise interpretation rather than pure listing.
Thank you for your comment. You're right that dividing it into subsections would make it easier to understand. However, this would complicate an already lengthy text, so we ask that you leave this text as is.
Discussion
- I suggest groupingthe relevant themes in the discussion.
(1) Lipophilic diversity,
(2) Organ-specific metabolism,
(3) Bioactivity linkages.
Thank you for your comment. We've restructured the discussion section based on your feedback.
- Please strengthen the link to SARS-CoV-2 protease inhibition with literature comparison (e.g., triterpenoids in other species).
Thank you for your comment. We've updated the discussion section with links to papers analyzing various groups of inhibitor substances.

Round 2
Reviewer 1 Report
Comments and Suggestions for Authors
I have reviewed the second revision of the manuscript. I am satisfied with the changes made, which have improved the quality of the presentation. The manuscript is well-prepared and ready for publication.
Reviewer 2 Report
Comments and Suggestions for Authors
No further changes are required.